# Prognostic Role of Post-Induction Fecal Calprotectin Levels in Patients with Inflammatory Bowel Disease Treated with Biological Therapies

**DOI:** 10.3390/biomedicines10092305

**Published:** 2022-09-16

**Authors:** Antonio Facciorusso, Daryl Ramai, Cristina Ricciardelli, Rosa Paolillo, Marcello Maida, Saurabh Chandan, Babu P. Mohan, Viktor Domislovic, Rodolfo Sacco

**Affiliations:** 1Gastroenterology Unit, Department of Surgical and Medical Sciences, University of Foggia, 71122 Foggia, Italy; 2Division of Gastroenterology, Hepatology and Nutrition, School of Medicine, University of Utah, Salt Lake City, UT 84112, USA; 3Gastroenterology and Endoscopy Unit, S. Elia-Raimondi Hospital, 93100 Caltanissetta, Italy; 4Gastroenterology Unit, CHI Health Creighton University Medical Center, Omaha, NE 68131, USA; 5Department of Gastroenterology and Hepatology, University Hospital Centre Zagreb, 10000 Zagreb, Croatia

**Keywords:** IBD, mucosal healing, response, sensitivity, accuracy

## Abstract

Background: There is currently scarce knowledge about markers of early therapeutic response in patients with inflammatory bowel disease (IBD) treated with biologics. The aim of this study was to evaluate the role of fecal calprotectin (FC) as an early predictor of mucosal healing and clinical remission. Methods: Data from a multicenter series of 172 IBD patients treated with biologics between 2017 and 2020 were analyzed. Treatment outcomes were mucosal healing and clinical remission assessed at 2 years. FC levels were assessed at 14 weeks (post-induction), at 6 months, and yearly. The receiver operating characteristic (ROC) curve analysis was performed to calculate the best cut-off in % change of FC levels between post-induction and baseline predicting treatment outcomes. Sensitivity, specificity, and accuracy for several post-induction FC cut-off points were also calculated. Results: At 2 years, mucosal healing was noted in 77 patients (44.7%), of whom were 41 Crohn’s disease (CD) and 36 ulcerative colitis (UC) patients, whereas 106 patients experienced clinical remission (61.6%), of whom were 59 CD and 47 UC patients. Both baseline and post-induction FC levels were significantly higher in non-responders as compared to responders. On the other hand, FC decrease was less pronounced in non-responders. Similar results were observed in all subgroups, namely according to disease (CD vs. UC), or treatment used (TNF-inhibitors vs. vedolizumab). The best cut-off points were −86% in % change in FC levels to predict mucosal healing and −83% for clinical remission. Conclusions: The current study suggests a predictive role of post-induction FC assessment to predict treatment response in IBD patients treated with biologics.

## 1. Introduction

In recent years, treatment targets in inflammatory bowel disease (IBD) have been directed towards deep endoscopic and histologic remission in order to prevent relapses and complications [1].

The anti–tumor necrosis factor (TNF), monoclonal antibodies infliximab (IFX), adalimumab (ADA), and golimumab (GOL) have greatly improved the management of both patients with ulcerative colitis (UC) and Crohn’s disease (CD); furthermore, other therapeutical options are currently available, such as vedolizumab (VDZ), that is an anti-integrin monoclonal antibody able to prevent specific T lymphocytes from endothelial adherence and migration in the bowel mucosa [2].

However, a proportion of patients still experience poor or no response to these treatments. Therefore, early prediction of response to biological therapies plays a pivotal role in order to optimize treatment and eventually switch to alternative therapies. In this regard, the identification of a reliable and non-invasive biomarker would allow to optimize the management of these patients, particularly through serial assessments at different points in time.

Among biomarkers, fecal calprotectin (FC) was shown to be strongly correlated with both endoscopic and clinical remission in IBD patients [3,4]. However, static cross-sectional assessment of FC levels at baseline presents several limitations including the ability to customize treatment strategy based on the clinical course of patients under treatment. On the other hand, the identification of a predictive role of early response to treatment evaluated by means of post-induction FC levels assessment could be helpful to tailor individual treatment.

A reliable biomarker should demonstrate robust association with long-term treatment outcomes, including surrogate endpoints (clinical and endoscopic remission). In this regard, FC assessment likely represents the best test as changes in FC levels could be detected before the onset of any visible changes in disease activity [5,6,7]. This suggests that FC assessment may have a role as a dynamic measure in a treat-to-target paradigm.

Previous studies demonstrated an association between week 8 FC concentration ≤250 μg/g and endoscopic response at week 16 in IBD patients treated with vedolizumab [8]. However, the prognostic value of early changes in FC concentration for long-term outcomes is still unclear as it was suggested only in small series [9] or in post-hoc analyses of previous trials [10,11] without a clear definition of the kinetics of this biomarker over time.

We aim to evaluate the association of post-induction change in FC levels with long-term treatment outcomes in IBD patients treated with biologic drugs (TNFi or VDZ).

## 2. Materials and Methods

### 2.1. Patients

Data were retrospectively retrieved from a prospectively multicenter database of consecutive adult IBD patients who underwent biological treatment with IFX, ADA, GOL, or VDZ between Jan 2017 and Oct 2020.

Data of the enrolled patients were collected from the day of the first dose of biological treatment and subsequently evaluated through clinical visits every 8 weeks for 2 years.

Indication for biologic treatment was active IBD defined as Crohn’s Disease Endoscopic Index of Severity (CDEIS) score >6 in the case of CD patients or Mayo Clinic Score (MCS) of 6 to 12, with endoscopic subscore ≥2 in the case of UC patients. Patients were also treated with biologics in the case of an inadequate response to steroids (resistance defined as active disease despite prednisolone of up to 40 mg/d over a period of 2 weeks; dependence as inability to reduce prednisolone below 10 mg/d within 3 months or relapse within 3 months of stopping steroids; intolerance defined as steroid-related side-effects leading to steroid discontinuation). Some patients also had a history of non-response or intolerance to immunosuppressants (thiopurines or methotrexate).

Primary non-responders to TNFi or VDZ, defined as a decrease in Mayo Score ≤2 in UC patients or a lack of clinical improvement at week 8, were excluded. Other exclusion criteria were the inability to collect fecal samples, pregnancy, infectious enterocolitis, colorectal cancer, indeterminate colitis, and contraindications for anti-TNF therapy. Approbation for the retrospective analysis of de-identified clinical data was obtained.

### 2.2. Treatment and Follow-Up

The treatment regimen was as follows: 5 mg/kg i.v. at weeks 0, 2, and 6 and then every 8 weeks for IFX; 160 mg at week 0, 80 mg at week 2, and 40 mg subcutaneously starting from week 4 for ADA; 200 mg at week 0, 100 mg at week 2, and then 50 mg or 100 mg (if patient’s weight was >80 kg) subcutaneously every 4 weeks for GOL; and 300 mg at weeks 0, 2, and 6 and then every 8 weeks for VDZ.

Treatment could be escalated in patients with worsening disease where testing for antidrug antibodies was performed.

Baseline demographical and clinical data were recorded, including disease extension, concomitant corticosteroid/immunosuppressant treatment, FC, MCS, and MES/CDEIS.

FC was assessed at baseline, at 14 weeks (after treatment induction), at 6 months, and yearly using the ELISA Bühlmann fCAL Turbo assay (Bühlmann Laboratories AG, Schönenbuch, Switzerland).

Colonoscopy was performed at baseline, after 1 year, and after 2 years for endoscopic disease assessment and therapeutic response.

### 2.3. Treatment Outcomes

Clinical remission was defined as an MCS ≤ 2 with no subscore > 1. Mucosal healing was defined as a MES of 0 or 1. These clinical and endoscopic outcomes were assessed at 1 year and at 2 years after treatment.

### 2.4. Statistical Analysis

Categorical variables were described as frequencies and percentages while continuous variables were defined as median and interquartile range (IQR).

The receiver operating characteristic (ROC) curve analysis (using a non-parametric test) was performed to calculate the best cut-off in delta (% change) FC between post-induction and baseline and absolute FC post-induction levels predicting either mucosal healing or clinical remission. Sensitivity, specificity, and accuracy for both treatment outcomes testing different post-induction FC cut-off points, namely 50, 100, and 250 mg/kg, were also calculated as described elsewhere [12,13]. A sub-analysis restricted only to patients with baseline FC levels > 250 mg/kg was performed.

Comparison between point estimates of FC levels at different time points was performed by means of Kruskal–Wallis test. The analysis was performed using R Statistical Software 4.2.1 version (Foundation for Statistical Computing, Vienna, Austria), and significance was established at the 0.05 level (two-sided).

## 3. Results

### 3.1. Baseline Characteristics of the Patients

Clinical and demographic characteristics of 172 patients are summarized in Table 1. Median age was 43 years (IQR 38–54) and 102 patients (59.3%) were male.

Previous immunomodulator therapy was registered in 66 subjects (38.3%) whereas previous TNFi treatment was present in 25 patients (14.5%). Out of the 172 included patients, 99 (57.5%) presented CD and 73 (42.5%) presented UC. In CD patients, disease extension was L1 in 32 patients (32.3%), L2 in 6 patients (5.9%), and L3 in 61 patients (61.8%), whereas disease behavior was classified as B1, B2, and B3 in 63 (63.6%), 34 (34.3%), and 2 (2.1%) patients, respectively.

Disease extension in UC patients was E1 in 12 patients (16.4%), E2 in 17 (23.2%) patients, and E3 in 44 (60.4%) patients. Median baseline FC level was 452 (242–570) mg/kg. Concomitant immunosuppression was present in 79 patients (45.9%). In the study population, 31 patients (18%) used IFX, 62 (36%) ADA, 36 (20.9%) GOL, and 43 (25.1%) VDZ.

### 3.2. Correlation between Post-Induction FC Levels and Treatment Outcomes

At 2 years, mucosal healing was observed in 77 patients (44.7%), of which 41 CD and 36 UC patients. Among these patients, 58 were under TNFi treatment and 19 used VDZ. Similarly, 106 patients experienced clinical remission (61.6%), of which 59 CD and 47 UC patients. Clinical remission was observed in 80 out of 129 patients under TNFi (62%) and 26 out of 43 patients under VDZ treatment (60.4%).

As reported in Table 2, both baseline FC levels and FC levels assessed at 14 weeks (post-induction) were significantly higher in non-responders as compared to responders (*p* < 0.001).

In particular, as depicted in Figure 1, FC decreased from 242 (113–432) mg/kg to 42 (13–132) mg/kg and from 313 (216–451) mg/kg to 53 (16–151) mg/kg in patients who experienced mucosal healing and clinical remission at 2 years, respectively.

On the other hand, FC decrease was less pronounced in non-responders, specifically from 856 (459–1321) mg/kg to 756 (359–1212) mg/kg and from 1057 (655–1423) mg/kg to 785 (351–1004) mg/kg considering mucosal healing or clinical remission, respectively. Similar results were observed in all the subgroups, namely according to disease (CD vs. UC), or treatment used (TNFi vs. VDZ; Table 2).

Considering only the 157 patients with baseline FC > 250 mg/kg, FC levels decreased from 389 (252–459) at baseline to 48 (21–189) at week 14 in patients with mucosal healing and from 915 (569–1443) at baseline to 813 (387–1349) at week 14 in those who did not experience mucosal healing. Likewise, FC levels decreased from 398 (254–502) at baseline to 59 (23–172) at week 14 in patients with clinical remission and from 1118 (741–1811) at baseline to 812 (518–1324) at week 14 in those subjects who did not achieve clinical remission.

To calculate the best cutoff of delta FC between week 14 and baseline related to treatment outcomes, an ROC analysis was performed (Figure 2). The best cut-off points were −86% in delta FC to predict mucosal healing and −83% for clinical remission, with an area under the curve (AUC) of 0.81 and 0.78, respectively.

When selecting a post-induction FC cut-off point of 50 mg/kg, accuracy, sensitivity, and specificity were 88.3% (82.6–92.7%), 75.3% (64.2–84.4%), and 98.9% (94.3–99.9%) for mucosal healing and 69.2% (61.7–76%), 52.8% (42.9–62.6%), and 95.4% (87.3–99%) for clinical remission, respectively. Increasing the FC cut-off value to 100 mg/kg led to an improvement in accuracy (86%, 79.9–90.8% for mucosal healing and 80.8%, 74.1–86.4% for clinical remission) and sensitivity (83.1%, 72.8–90.7% for mucosal healing and 69.8%, 60.1–78.3% for clinical remission), with a decrease in specificity (88.4%, 80.2–94.1%) to detect mucosal healing (Table 3).

On the other hand, specificity to detect clinical remission remained high (98.5%, 91.8–99.7%). Finally, selecting a cut-off value of 250 mg/kg determined accuracy, sensitivity, and specificity of 90.7% (85.3–94.6%), 98.7% (93–99.9%), and 84.2% (75.3–91%) for mucosal healing and 90.1% (84.6–94.1%), 84.9% (76.6–91.1%), 98.5% (91.8–99.9%) for clinical remission. ROC analysis using absolute post-induction FC levels reported at Figure 3 showed that 141 mg/kg and 135 mg/kg were the best cut-off absolute values for predicting mucosal healing and clinical remission, respectively.

## 4. Discussion

Integration of repeated measurements of disease activity into treatment algorithms to make time-dependent treatment adjustments represents a pressing need in the management of IBD patients [14]. However, the need for frequent endoscopies has important convenience, cost, and access limitations, hence the need for a reliable non-invasive biomarker such as FC. Furthermore, it is currently unclear how long the mucosa takes to demonstrate macroscopic (endoscopy) and microscopic (histology) changes in response to treatment of the underlying disease. To this end, a biomarker might be able to detect an early response to treatment and inform clinical decisions.

Calprotectin is a 36 kDa calcium and zinc binding protein representing about 60% of soluble proteins of the cytoplasm of granulocytes [15]. It is a heat and proteolysis resistant heterocomplex of S100A8 and S100A9 consisting of two heavy (14 kDa) e 1 light (8 kDa) chains, each binding 2 Ca^2+^. Functions of calprotectin include competitive inhibition of zinc-dependent enzymes, potential biostatic activity against microbes through chelation of zinc ions, apoptosis induction in malignant cells, and regulation of the inflammatory process [15].

FC represents an acute marker of degranulation which is correlated with neutrophil infiltration into the mucosa. FC could be an ideal marker for early response because neutrophilic infiltration into the mucosa is a preliminary and major driver of the immune process in active IBD [16].

To the best of our knowledge, our study represents the first series to report several consecutive serial assessments of FC levels in a large multicenter cohort of IBD patients treated with biologics. While previous studies demonstrated a predictive role of early change in FC levels, the assessment of this biomarker was limited only to baseline and after treatment induction whereas longer-term evaluations were lacking [9,10,11,17].

Our series confirms the favorable results of biological treatments in the management of active IBD, with rates of mucosal healing and clinical remission at 2 years of 44.7% and 61.6%.

In our study, we observed a significant difference between FC levels assessed both at baseline and after induction (14 weeks) between responders and non-responders. A substantial decrease in FC levels was noted at 14 weeks in patients who finally experienced mucosal healing and clinical remission.

Previous studies found that FC levels <100 mg/kg after the induction of TNFi therapy predicts clinical remission at 1 year and a cut-off of 139 mg/kg was identified as being able to predict a risk of clinically active disease after 1 year, with a sensitivity of 72% and a specificity of 80% [18]. Our study was focused on the delta (% change) in FC levels rather than a post-induction absolute value as this would be influenced by baseline FC levels. Therefore, we do believe a relative measure such as the percentage change in FC levels might be more reliable and an objective tool in this setting.

Evidence on the prognostic role of early FC change while considering mucosal healing as a primary outcome is still limited. A Belgian prospective study showed that FC assessed at week 10 had a very good correlation with mucosal healing evaluated at the same time point in UC patients treated with IFX [19]. A post-hoc analysis of two RCTs found a significant predictive role for mucosal healing of post-induction FC level using 250 mg/kg as a cut-off point [11] whereas evidence from real world data are limited mainly to small single-center series [9,20]. In this regard, our study represents a large series with multiple subgroup analyses aimed at assessing the predictive role of early FC changes in different subsets of patients. Moreover, our study assessed treatment outcomes up to 2 years.

Additionally, we observed that with the increase in FC cut-off levels, there is an overall increase in predictive accuracy, hence we could suggest a cut-off of 250 mg/kg in post-induction FC levels for both clinical remission and mucosal healing. This finding is in line with previous reports [11]. Of note, AUC of delta FC level was 0.81 and 0.78 for mucosal healing and clinical remission, respectively, thus strongly supporting the use of this biomarker.

Our results could be applied to clinical practice for detecting non-responders that should be switched to other therapeutical options. This could be particularly important in several subsets of patients, for example oncological patients who experience IBD flares induced by newer immune checkpoint inhibitors [21,22].

Different methods can be used for the quantitative assessment of FC, most of them based on the enzyme-linked immunosorbent assay (ELISA); chemiluminescence immunoassays (CLIA), fluoro enzyme immunoassays (FEIA), and particle enhanced turbidimetric immunoassays (PETIA) were also introduced. In our study, the ELISA Bühlmann fCAL Turbo assay was used. In fact, while other assays provide a semi-quantitative result, ELISA is a quantitative method on a microtiter plate. Of note, no significant differences were found in terms of diagnostic accuracy among different assays testing FC [23].

FC is currently a valuable and widely used fecal marker for differentiating between IBD and functional disorders and due to the high negative predictive value, it is highly accurate in ruling out intestinal inflammation both in primary and secondary care. However, it should be noted that this marker is inflammatory but not disease-specific, hence the need for other diagnostic tests and biomarkers for the exact diagnosis of CD or UC [24].

This study presents several limitations. First, the retrospective nature could have led to some forms of selection bias. Second, FC levels can vary day-to-day, by the time of day, and even within the same bowel movement [25]. Moreover, they can be influenced by some drugs such as non-steroid anti-inflammatory drugs (NSAIDs) [26]. However, our study represents the first series reporting several consecutive assessments of FC levels over time up to 2 years.

Despite these limitations, we demonstrated an early predictive value of FC in IBD patients treated with several biological therapies. This finding encourages the wide use of FC to obtain an early assessment and tailor treatment strategies for patients.

## 5. Conclusions

The current study suggests a predictive role of post-induction FC assessment to predict treatment response in IBD patients treated with biologics. Patients with higher post-induction levels of FC should be closely monitored, anticipating colonoscopy to evaluate endoscopic remission or detect IBD-related mucosal lesions, and to optimize (or switch) their treatment.

## Figures and Tables

**Figure 1 biomedicines-10-02305-f001:**
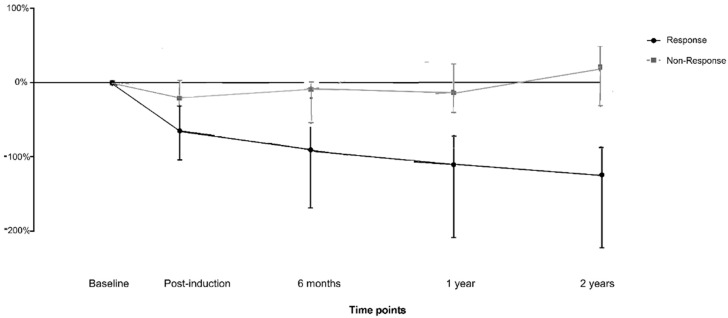
Kinetics of fecal calprotectin levels in responders and non-responders at different consecutive time-points.

**Figure 2 biomedicines-10-02305-f002:**
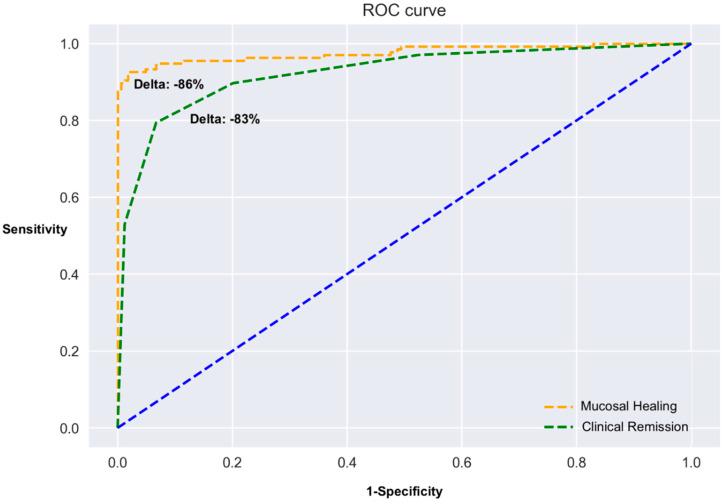
Receiver operating characteristics (ROC) curve assessing the predictive performance of delta (% change) fecal calprotectin after induction with respect to baseline for mucosal healing and clinical remission.

**Figure 3 biomedicines-10-02305-f003:**
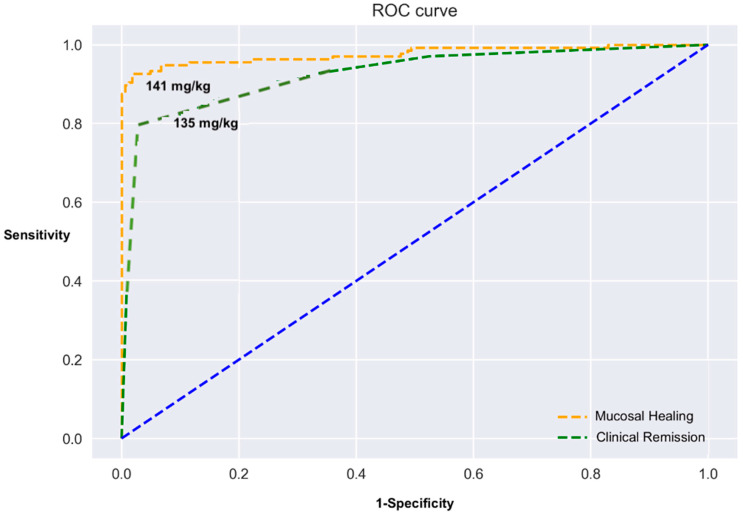
Receiver operating characteristics (ROC) curve assessing the predictive performance of absolute fecal calprotectin levels after induction for mucosal healing and clinical remission.

**Table 1 biomedicines-10-02305-t001:** Patients’ baseline characteristics.

Variable	Total (172 Patients)
Age (years)	43 (38–54)
Gender (Male)	102 (59.3%)
Previous immunomodulator	66 (38.3%)
Previous TNFi treatment	25 (14.5%)
Disease	
Crohn disease	99 (57.5%)
Ulcerative colitis	73 (42.5%)
Disease extension Crohn disease	
L1	32 (32.3%)
L2	6 (5.9%)
L3	61 (61.8%)
Disease behavior Crohn disease	
B1	63 (63.6%)
B2	34 (34.3%)
B3	2 (2.1%)
Disease extension Ulcerative colitis	
E1	12 (16.4%)
E2	17 (23.2%)
E3	44 (60.4%)
Fecal calprotectin (mg/kg)	452 (242–570)
Concomitant immunosuppression	79 (45.9%)
Drug	
Infliximab	31 (18%)
Adalimumab	62 (36%)
Golimumab	36 (20.9%)
Vedolizumab	43 (25.1%)

**Table 2 biomedicines-10-02305-t002:** Fecal calprotectin values at baseline and after 14 weeks of treatment according to drug used and treatment response.

Variable		Mucosal Healing		Clinical Remission	
	Overall	Responders	Non-Responders	Responders	Non-Responders
Total (172 pts)		77 (44.7%)	95 (55.3%)	106 (61.6%)	66 (38.4%)
Baseline FC	452 (242–570)	242 (113–432)	856 (459–1321)	313 (216–451)	1057 (655–1423)
W14 FC	112 (89–404)	42 (13–132)	756 (359–1212)	53 (16–151)	785 (351–1004)
Crohn disease (99 pts)		41 (41.4%)	58 (58.6%)	59 (59.5%)	40 (40.5%)
Baseline FC	515 (311–723)	341 (134–451)	952 (445–1132)	322 (243–631)	1251 (658–1521)
W14 FC	143 (102–512)	48 (22–143)	857 (443–1239)	59 (13–188)	897 (544–1323)
Ulcerative colitis (73 pts)		36 (49.3%)	37 (50.7%)	47 (64.3%)	26 (35.7%)
Baseline FC	412 (225–513)	227 (115–399)	665 (449–1213)	319 (189–555)	948 (632–1320)
W14 FC	105 (78–439)	38 (11–129)	649 (459–1118)	51 (18–198)	714 (343–1017)
TNFi (129 pts)		58 (44.9%)	71 (55.1%)	80 (62%)	49 (38%)
Baseline FC	463 (234–564)	276 (134–531)	889 (434–1411)	334 (211–576)	1059 (752–1388)
W14 FC	108 (85–398)	40 (12–154)	715 (559–1565)	49 (16–187)	798 (413–1404)
Vedolilzumab (43 pts)		19 (44.1%)	24 (55.9%)	26 (60.4%)	17 (39.6%)
Baseline FC	435 (258–581)	255 (191–513)	834 (423–1311)	319 (203–443)	1023 (603–1521)
W14 FC	143 (93–415)	38 (18–165)	787 (355–1401)	49 (18–139)	879 (543–1433)

Abbreviations: FC, fecal calprotectin; FC levels were expressed in terms of mg/kg.

**Table 3 biomedicines-10-02305-t003:** Predictive performance for treatment outcomes of post-induction fecal calprotectin levels based on different cut-off points.

Cut-Off	Mucosal Healing				Clinical Remission			
	Responders	Accuracy	Sensitivity	Specificity	Responders	Accuracy	Sensitivity	Specificity
<50 mg/kg		88.3% (82.6–92.7%)	75.3% (64.2–84.4%)	98.9% (94.3–99.9%)		69.2% (61.7–76%)	52.8% (42.9–62.6%)	95.4% (87.3–99%)
Yes (59 pts)	58				56			
No (113 pts)	19				50			
<100 mg/kg		86% (79.9–90.8%)	83.1% (72.8–90.7%)	88.4% (80.2–94.1%)		80.8% (74.1–86.4%)	69.8% (60.1–78.3%)	98.5% (91.8–99.7%)
Yes (75 pts)	64				74			
No (97 pts)	13				32			
<250 mg/kg		90.7% (85.3–94.6%)	98.7% (93–99.9%)	84.2% (75.3–91%)		90.1% (84.6–94.1%)	84.9% (76.6–91.1%)	98.5% (91.8–99.9%)
Yes (91 pts)	76				90			
No (81 pts)	1				16			

## Data Availability

The data presented in this study are available upon request from the corresponding author.

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
