# Peer review of "Prognostic Role of Post-Induction Fecal Calprotectin Levels in Patients with Inflammatory Bowel Disease Treated with Biological Therapies"

_biomedicines, 2022, doi:10.3390/biomedicines10092305_

Round 1

Reviewer 1 Report

There are many reviews on fecal calprotectin in IBS with detail critical review.Only edge your study has that you have included scores to measures in study.

Add sentences about fecal Calprotectin.

Methods used to measure fecal calprotectin.

Authors should lay more stress that this marker is inflammation but not disease specific.

Author Response

There are many reviews on fecal calprotectin in IBS with detail critical review. Only edge your study has that you have included scores to measures in study.

RE: We appreciate the comment raised by the reviewer. Just to specify, our paper is not a review on fecal calprotectin but an original series reporting the kinetics of this marker over time (along with the predictive value of this kinetics for final response) in the two different groups of responders vs non-responders. In this regard, our series represents one of the largest so far with this specific aim.

Add sentences about fecal Calprotectin.

RE: Further sentences on fecal calprotectin were added to the Discussion with a new reference (ref 15).

Methods used to measure fecal calprotectin.

RE: The assay used in our study to measure FC was specified in the Material and Methods section (Page 3). However, based on the reviewer’s comment, we added another sentence on this topic (along with a new reference) in the Discussion.

Authors should lay more stress that this marker is inflammation but not disease specific.

RE: This aspect was further commented in the Discussion and a new reference on this topic was added.

Reviewer 2 Report

This is an interesting study which seeks to determine a cut off of the decrease in fecal calprotectin under the effect of short-term biotherapy, week 14. This is a retrospective cohort study.

We would like to have in table 2, a column of pooled results for all IBD, and not only in responders or non-responders and this for the two endpoints. Indeed, the usual threshold of 250 µg/g being usually used, we would like all the patients included in this study to have a threshold > 250 on inclusion and not for the average or the median of the patients to be 250. answer seems to be given line 148 and therefore it is a problem for the study that the lower limit of the confidence interval is at 113 and 216 at inclusion.

If the search for a threshold in percentage of reduction is interesting, we would also like to have the result of the cut off in value of fecal calprotectin. From the point of view of the statistical choice to express a cut off as a percentage reduction, it would be useful for the authors to justify the choice of this unit from a statistical point of view because it seems to me that the ROC curves are mainly used with units dosage but not percentages. To be checked especially since the authors still use absolute values ​​to vary their calculation (line 167). In short, we do not understand the main criterion in percentage and use of the variation of absolute value of the level of fecal calprotectin to choose the right threshold.

Finally, there are some additional references by searching for "trough levels, fecal calprotectin, early".

Author Response

This is an interesting study which seeks to determine a cut off of the decrease in fecal calprotectin under the effect of short-term biotherapy, week 14. This is a retrospective cohort study.

We would like to have in table 2, a column of pooled results for all IBD, and not only in responders or non-responders and this for the two endpoints. Indeed, the usual threshold of 250 µg/g being usually used, we would like all the patients included in this study to have a threshold > 250 on inclusion and not for the average or the median of the patients to be 250. answer seems to be given line 148 and therefore it is a problem for the study that the lower limit of the confidence interval is at 113 and 216 at inclusion.

RE: We added data concerning the whole cohort to Table 2, as suggested.

We aimed to include in our study all consecutive IBD patients undergoing biological treatment in the recruiting period in order to overcome any risks of selection bias. In fact, we aimed to test the predictive value of FC decrease at W14 in all IBD patients. However, based on the valuable reviewer’s comment, we added a sub-analysis restricted only to patients with baseline FC >250 mg/kg and reported the results at page 5 (Results section).

Of note, at line 148 it is reported the lower confidence intervals of patients experiencing response, not all the patients (hence the lower values). The confidence interval concerning the whole cohort is reported in Table 1 and it was 452 (242-570) mg/kg. We have now specified in the text that 157 patients (so the vast majority of patients) presented with baseline FC levels >250 mg/kg.

If the search for a threshold in percentage of reduction is interesting, we would also like to have the result of the cut off in value of fecal calprotectin. From the point of view of the statistical choice to express a cut off as a percentage reduction, it would be useful for the authors to justify the choice of this unit from a statistical point of view because it seems to me that the ROC curves are mainly used with units dosage but not percentages. To be checked especially since the authors still use absolute values ​​to vary their calculation (line 167). In short, we do not understand the main criterion in percentage and use of the variation of absolute value of the level of fecal calprotectin to choose the right threshold.

RE: The reason why the main analysis was performed on a “delta” (a difference) is due to the fact that this value is more robust and not highly dependent on baseline FC levels such as the absolute cut-off point. For example, if we have a baseline FC level of 400, the absolute FC level at 14 weeks will be (for example) 200. If we have a baseline FC level of 800, the level will be likely higher than 200 but this is due just to the higher “starting point” (baseline level). With the delta (percentage decrease) this potential bias could be at least partially overcome. However, following the reviewer’s suggestion, we added a further ROC curve (Figure 3) based on absolute values.

Finally, there are some additional references by searching for "trough levels, fecal calprotectin, early".

RE: Based also on reviewer 1’s suggestions, we added 3 further references mainly on FC.

Round 2

Reviewer 2 Report

good corrections that fit to me, thank you